# Damoctocog Alfa Pegol for Hemophilia A Prophylaxis: An Italian Multicenter Survey

**DOI:** 10.3390/ph16091195

**Published:** 2023-08-22

**Authors:** Ezio Zanon

**Affiliations:** Hemophilia Center, General Internal Medicine, Department of Medicine Padua University Hospital, 35128 Padua, Italy; ezio.zanon@unipd.it; Tel.: +39-04-9821-2666

**Keywords:** Hemophilia A, coagulation factor VIII deficiency, prophylaxis, damoctocog alfa pegol

## Abstract

Hemophilia A is characterized by a deficiency of clotting factor VIII (FVIII) requiring lifelong prophylactic treatment, typically with recombinant FVIII. In recent years, drugs with extended half-lives have become available, including damoctocog alfa pegol (Bayer S.p.A.). The clinical efficacy and safety of damoctocog alfa pegol were demonstrated in the PROTECT VIII phase II/III development program. To assess the physicians’ experience and to collect data on prophylactic treatment with damoctocog alfa pegol, a monitoring survey was carried out among 15 Italian hemophilia centers. A total of 149 patients on treatment with damoctocog alfa pegol for at least 6 months were considered. Zero bleeds were reported in 75% of patients treated with damoctocog alfa pegol in the last 6 months; zero hemarthroses were reported in 82% of the same patients. Overall, 86% of patients with damoctocog alfa pegol reduced their monthly infusions. The estimated average reduction in IU/kg during prophylaxis with damoctocog alfa pegol, both monthly and annually, was around 17.5%. All involved clinicians were satisfied with damoctocog alfa pegol. Survey results suggest that damoctocog alfa pegol reduced the number of weekly infusions, annual bleeding, and joint bleeding rate in the majority of patients, improving joint health and patients’ quality of life.

## 1. Introduction

Hemophilia A is a rare bleeding disorder known since ancient times. However, the perception of the disease, the treatments, the expectations and quality of life, and the needs of persons with hemophilia (PWH) have changed over the ages.

Although hints of people with excessive bleeding can be found in the writings of Hippocrates of Kos, the first true references to a bleeding condition suggestive of hemophilia date back to the 2nd century AD in the Babylonian Talmud, while a detailed description of the disease can be found in the Kitab al-Tasrif written by the Arab physician, Albucasis [1,2,3,4]. Only after 1828, however, this hemorrhagic disorder was named hemophilia by Friedrich Hopff and Johann Lukas Schönleinm, which we all know today [5,6]. Though the disease was known throughout all these centuries, the treatments were not, and the life expectancy of the patients was very low. A Swedish review published by Larsson in 1985 [7] showed how the life expectancy of patients with severe hemophilia in the years 1830–1920 was only 11 years, increased to 23 years in the period 1921–1940, to 28 years between 1941 and 1960, until reaching the 57 years in the 20 years 1961–1980. In this period, patients were looking for treatments to prevent them from bleeding, having severe hemorrhages (e.g., intracranial hemorrhage), and dying at a young age. Although the first blood transfusions to help people suffering from acute bleeding date back to the mid-19th century [8], only thanks to the discovery of blood groups by Karl Landstainer in 1900 [9] these treatments could be used more effectively. In 1935, there was talk of a coagulation-promoting substance obtained from human plasma that, administered intramuscularly or intravenously, could reduce the coagulation time of hemophilia patients [10], which was followed by the discovery of cryoprecipitate in 1965 [11] and desmopressin in 1977 [12]. All this took place in Sweden [13], while there was talk of prophylaxis in treating hemophilia, and patients expressed needs hitherto unknown, such as those of being able to work and study like the rest of the population. In the 1980s, the first plasma-derived FVIII concentrates were produced and distributed to help implement prophylaxis in hemophilic patients, thereby improving their quality of life. Unfortunately, the great euphoria given by the discovery of ever-new treatments for hemophilia is accompanied in the 90s by the great pandemic of HIV and HCV, which claims victims and causes the onset of chronic diseases [14]. Patients, therefore, begin to ask for greater safety of FVIII concentrates, which is subsequently achieved thanks to a more awaited screening of blood donors, pasteurization, and viral inactivation of plasma-derived products [14,15,16]. Clinical research, therefore, helps patients’ new needs, and in 1991 the first FVIII of recombinant origin was licensed [17]. Viruses are, therefore, no longer a problem for hemophilia patients. Therefore, the major risk is represented by developing inhibitors against FVIII and its consequences while patients perform new requests.

In the second decade of the 2000s, the unmet needs changed; patients first asked for a reduction in the number of infusions, followed by a documented efficacy of the products and their safety, easier handling, and improved quality of life.

In recent years, alongside the traditional therapy with standard half-life concentrates, both to respond to the needs of clinicians who require increasingly effective and safe treatments and to respond to these new patient needs, drugs with extended half-life (EHL), including damoctocog alfa pegol [18] were developed.

Hemophilia A is a rare disease, but it is currently established that its overall prevalence in the world is 17.1 cases per 100,000 male births, one-third of which concern the most severe form of the disease. The number of hemophilia patients worldwide is, therefore, estimated to be 1,125,000, of which one-third, 418,000, are expected to have severe hemophilia [19]. It is, therefore, necessary to have effective and easily available drugs to meet the needs of these patients.

Damoctocog alfa pegol (Bayer) is a recombinant B-domain deleted FVIII (BDD-rFVIII) to which a site-specific 60-kDa polyethylene glycol (PEG) molecule has been linked, extending the plasma half-life of the drug. Its use has been approved in patients with hemophilia A ≥12 years [20,21]. Damoctocog alfa pegol is administered intravenously at doses and timing derived from the PROTECT VIII trials [22]. The efficacy and safety of this drug, first demonstrated in pivotal studies, were recently confirmed by Reding et al. [23] in their post hoc analysis of the PROTECT VIII study. A sub-group of 34 patients aged ≥40 years with hemophilia A and one or more comorbidities was included in this analysis. The treatment was well-tolerated, no severe adverse events were reported, and the annualized bleeding rate was also reduced in the main and extension study, whose data included up to 7 years, thus supporting the use of damoctocog alfa pegol for long-term therapy [23]. Some minor side effects of the drug have been reported and described in the product data sheet [24]; among these, the most common are headache (13%), pyrexia (9%), cough (8%), vomiting (5%), injection site reactions (4%) and hypersensitivity (4%).

Given the favorable results of the pivotal and extension trials, clinicians should know the impact of these new drugs on daily clinical practice. Indeed, real-world studies, registries, and surveys are the tools currently available to obtain these answers. The HEM-POWR study, still ongoing, is a prospective, multicenter, non-interventional phase IV study that aims to evaluate the efficacy and safety of damoctocog alfa pegol in real-world treatment, with particular attention to joint health, pharmacokinetics (PK), and patient-reported outcomes [25].

In 2017, Lieuw [26] pointed out that the wide availability of products for treating hemophilia A could create a certain embarrassment for the clinician in choosing the most suitable treatment, but he also highlighted how this therapeutic abundance was an advantage for patients. In the same year, von Mackensen et al. [27] published an article clearly outlining the unmet needs of patients, prominent among which was the desire for drugs with a reduced number of infusions. Damoctocog alfa pegol seems to be able to reconcile these different needs: patients to have a safe and easy-to-use therapy and clinicians to have an effective and lasting therapy among the many available.

This survey aimed to evaluate the impact of damoctocog alfa pegol on the prophylaxis treatment of hemophilia A in real-life in a sample of Italian hemophilia centers.

## 2. Results

Overall, 15 hemophilia centers participated in this survey, managing 1721 patients with hemophilia A, an average of 115 patients per center. Of these, 984 (57%) were treated on prophylaxis with different FVIII concentrates; among these, damoctocog alfa pegol was the first treatment choice for 164 patients (17%).

Overall, 81 (49%) of patients were treated with damoctocog alfa pegol for over 12 months, 68 (42%) between 6 and 12 months; the remaining 15 (9%) received treatment for 6 or fewer months and were excluded from further investigations.

### 2.1. Switch to Damoctocog Alfa Pegol

During the scheduled visits of hemophilic patients, an evaluation of the bleeding episodes that occurred in the last period, an evaluation of the joint condition, and the quality of life are always carried out. Despite the prophylactic treatment, some patients may still present bleeding events due to imperfect hemostatic coverage or show the need to maintain a higher trough level suited to their lifestyle. Based on these observations and on the results of the PROTECT VIII program, which demonstrated the efficacy of damoctocog alfa pegol with a reduction in the number of infusions and with the possibility of maintaining higher plasma FVIII levels, a switch to this product was evaluated.

A total of 86% of patients currently receiving prophylactic treatment with damoctocog alfa pegol had previously been treated in prophylaxis with other FVIII concentrates, while 14% had previously been treated exclusively on demand; of these, 57% were affected by moderate hemophilia A. Almost all patients (83%) in the previous prophylactic treatment used standard half-life recombinant FVIII concentrates, more precisely 50% octocog alfa (Kovaltry), and 33% different standard half-life products (another octocog alfa, turoctocog alfa, moroctocog alfa, rFVIII-single chain, rhFVIII). Conversely, 3% of subjects were on previous therapy with other FVIII-EHL (efmoroctocog alfa, rurioctocog alfa pegol, turoctocog alfa pegol), 2% with plasma-derived FVIII concentrates; no data were available for the remaining 12%. Patients using standard half-life products were on prophylaxis at a standard dosage of 25–40 IU/kg three times a week, while patients in treatment with EHL products were treated with 50 IU/kg every 3–5 days.

The reduced number of bleeds, particularly joint bleeding, which led to a consequent improvement in the HEAD-US score in 40% of patients, is illustrated in Figure 1.

Zero bleeding was reported in 75% of patients treated with damoctocog alfa pegol in the last 6 months before this survey, while zero hemarthroses were reported in 82% of the same patients. The results were very different from those reported during the last 6 months of treatment with the previous drug, before the switch to damoctocog alfa pegol, in which patients who had not experienced bleeding or hemarthrosis were 38% and 50%, respectively.

A total of 57% of patients received an infusion every 5 days, and 10% an infusion every 7 days; 86% of patients with damoctocog alfa pegol reduced their monthly infusions from 11 (previous treatment) to six (−45% infusions with damoctocog alfa pegol). The reduction in the infusion frequency is shown in Figure 2.

However, the biweekly infusion regimen is more used in younger, active, and sporty patients, patients with synovitis or severe hemorrhagic phenotype, while the weekly regimen is used in older patients with a sedentary lifestyle and poor autonomy. The infusion regimen of choice in adults (18–65 years) was every 5 days.

The clinicians interviewed emphasized that the dosage and infusion timing choice depend on patient characteristics; Table 1 shows the three identified groups.

Based on the previously reported results concerning the choice of infusion in the hemophilia centers interviewed, it was possible to estimate the average monthly and annual factor consumption, comparing it with the previous treatment in the same patient (Table 2). In terms of dosage with damoctocog alfa pegol, there was a decrease in the average dosage per kilogram over 1 month compared with previous prophylaxis (average reduction of 69 IU/kg per month), a decrease that can reach approximately 57,000 IU per year for a hypothetical patient weighing 70 kg (−17.5%).

The average reduction in IU/kg during prophylaxis with damoctocog alfa pegol, both monthly and annually, was around 17.5%. The reduction in annual consumption could also positively impact economic sustainability. However, it was not possible to define this value in this manuscript, as there are varying price levels in different Italian regions.

### 2.2. Prophylaxis with Damoctocog Alfa Pegol: Clinicians’ Perception

In this survey, the clinicians involved were asked a few questions regarding their perception of the treatment with damoctocog alfa pegol; in particular, they were asked if they believed there had been a decrease in the number of bleeding events if the number of weekly infusions was reduced compared to the prior therapy, whether there was a reduction in weekly drug dosage, and whether patients were generally satisfied with this drug.

All the clinicians stated they were satisfied with damoctocog alfa pegol in terms of EHL, effectiveness, and the possibility of personalizing the patient’s therapy. The same clinicians positively rated the drug’s ability to reduce the number of infusions, improving treatment adherence and quality of life. The clinicians also felt that the patients treated with this FVIII-EHL concentrate were satisfied. Damoctocog alfa pegol was considered by the clinicians involved in the survey as the FVIII of first choice, compared with other EHLs, in many types of patients: sporty and active; those with chronic pain, arthropathy or synovitis; those with a severe bleeding phenotype; and those with poor autonomy or difficult venous access. Clinicians found that this drug could also provide significant benefits to patients with a sedentary lifestyle and pessimistic or disheartened psychological attitudes.

The aspects clinicians highlighted for further investigation regarding damoctocog alfa pegol are mainly related to the possibility of using it in patients younger than 12 years of age and reducing the PK variability given by different methods and laboratory reagents.

## 3. Discussion

Hemophilia has been known since ancient times as recorded in Greek and Arabic writings, but its treatment and management as we know it now is relatively recent. The discovery of blood groups made it possible to use blood transfusions to resolve acute bleeding by restoring blood volume and hemoglobin levels, but it did not help hemophiliac patients to prevent bleeding and its consequences. Only the advent of the first plasma-derived drugs to be used as a substitute concentrates for the deficient coagulation factors has therefore made it possible to intervene in a more targeted and early way in the treatment of hemophilic disease [1,2,14].

A milestone in the management of hemophilia is undoubtedly the 2007 study by Manco-Johnson et al. [28], which demonstrated without a shadow of a doubt how prophylaxis could prevent the onset of arthropathy in children and is, therefore, superior to demand treatment, widely used up to that moment.

Even today, prophylaxis with coagulation factor concentrates remains the gold standard of treatment for people with hemophilia, despite new non-replacement drugs already available or available soon. Numerous products can satisfy the requests of clinicians and patients; however, as reported by Lieuw [26], they can also create a certain doubt about the best treatment: a greater number of drugs and a greater difficulty of choice.

The latest World Federation of Hemophilia (WFH) guidelines show no differences between the various drugs, not suggesting a preference for using one over the other [29].

The statement was also recently reiterated by an expert panel of the Italian Association of Hemophilia Centers (AICE), which confirmed that it is impossible to establish one concentrate’s superiority over another, given the great heterogeneity of the products, their production, and processing. The therapeutic choice should be made according to patient characteristics, pharmacokinetics, needs, and concerns, applying particular attention to joint health and bleeding reduction [30].

Therefore, ad hoc studies, registries, and surveys are the only sources to draw the real-world data necessary to implement the best choice for patients. Using a survey as a source of information allows one to quickly obtain a lot of data regarding the efficacy and safety of a treatment. It allows researchers to photograph the reality of our Hemophilia Centers in an extremely simple way. Furthermore, the clinicians, always overwhelmed by many tasks, agree to participate in the surveys given the short time required to answer the questions. Our survey saw the participation of 15 clinicians from as many Hemophilia Centers and showed their motivations for choosing damoctocog alfa pegol, a site-specific PEGylated BDD-rFVIII, with an EHL [20], for the prophylactic therapy of patients with hemophilia A.

Adherence to therapy and lifelong optimization of prophylaxis are often hampered by several factors, clearly indicated by Núñez et al. [31]: the bleeding phenotype, which differs even among patients with the same degree of hemophilia as it depends on the genetic mutation involved, lifestyle, other concomitant diseases, etc.; joint status; physical activity; individual PK, which may be in favor of one drug over another, unlike for another similar patient; and other barriers, such as difficult venous access or poor patient education on prophylactic treatment. In this survey, damoctocog alfa pegol appeared to reduce the limitations listed above.

A very recent Dutch study by Versloot et al. [32], on 125 hemophilic patients aged between 6 and 49 years, with different degrees of hemophilia, all practicing sports of different intensity, demonstrated how the breakthrough bleedings were related to the level of the residual factor: in patients with levels of factor <10% there was an increased risk of bleeding which, on the other hand, was resulted not strictly correlated neither to the degree of hemophilia nor to the state of the joints nor the type of sport practiced.

Clinicians chose this drug for different types of patients; however, it emerged that a large number of patients treated with damoctocog alfa pegol were active and sporty and needed sustained hemostatic coverage to ensure an acceptable plasma level of FVIII to perform the physical activity safely while maintaining a small number of infusions.

Many of these assessed patients also had a severe bleeding phenotype. However, prophylactic use of damoctocog alfa pegol reduced the number of total and joint bleeds compared with the previous treatment, improving musculoskeletal function in nearly half of the patients. These real-world data support the results of the PROTECT VIII pivotal and extension studies [22,33].

Mancuso et al. [34], in the post hoc study published 2 years ago, clearly demonstrated the reduction of total and joint bleeding in patients who had switched from a standard half-life concentrate to damoctocog alfa pegol, even in the case of patients with documented severe bleeding phenotype.

Similar to the reports of Vashi et al. [35], who highlighted that, for the same efficacy, the annual consumption of damoctocog alfa pegol was 26.7% lower than that of turoctocog alfa pegol, the results of this survey also showed high efficacy of this drug with reduced consumption compared with previously used concentrates.

One of the major requests from patients has always been to reduce the number of infusions [25]; with damoctocog alfa pegol, this was achieved in over 85% of cases. A reduction in the number of infusions is usually associated with better adherence to treatment and improved quality of life; in the case of damoctocog alfa pegol, the clinicians interviewed emphasized that one of the strengths of this drug is primarily its ability to establish effective personalized therapy, improving treatment adherence, and quality of life.

Adherence to therapy is of paramount importance in reducing joint and total bleeding. Mokhatar et al. [36] analyzed data from the “Validated Haemophilia Regimen Treatment Adherence Scale–Prophylaxis (VERITAS-Pro)” questionnaires filled out by 126 with hemophilia A and B. They reported a significant difference in the mean annualized bleeding rate between the adherent and nonadherent subjects.

All the clinicians interviewed were satisfied with damoctocog alfa pegol, and patients treated with this drug expressed a positive vision of their future. As reported by Swaminathan et al. [37] in their letter to the editor, these responses confirm what data collected at 149 US hemophilia centers showed about how comfortable clinicians were using EHL drugs, often based on patient preferences.

A 2018 European consensus [38] established that the patients who could have benefited most from treatment with long-acting drugs are those who have poor adherence to therapy, those with severe bleeding phenotype, those who are very active, who practice sports, or those with difficult venous accesses, in summary, those that our survey demonstrated to be the most suitable for treatment with damoctocog alfa pegol according to the clinicians interviewed.

## 4. Materials and Methods

This survey involved a significant sample of Italian hemophilia centers with three or more patients with hemophilia A without inhibitors on prophylactic treatment with damoctocog alfa pegol (Bayer Italia S.p.A., Milan, Italy) for at least 6 months. The survey is divided into two sections: (1) the collection of cases and information regarding practical experience with damoctocog alfa pegol; (2) the collection of aggregated data on all patients on prophylactic treatment with damoctocog alfa pegol for at least 6 months.

The recommended dosage of damoctocog alfa pegol is 30–40 IU/kg two-times a week or 45–60 IU/kg every 5 days or 60 IU/kg every 7 days.

### Patients Treated with Damoctocog Alfa Pegol: Characteristics

In total, 83% of the assessed patients had severe hemophilia A, 15% moderate, and only 2% mild. The patients treated in prophylaxis with damoctocog alfa pegol were mainly young, over 50% under age 40 years, employed in work (58%) or schooling (24%), and with a medium (30%) to high (63%) degree of autonomy. Previous cardiovascular diseases, mainly hypertension under medication control, were reported in 53% of cases. The patients’ perceptions regarding their condition and chances for future improvement were positive; only 7% were disheartened.

Of all patients treated with damoctocog alfa pegol, 107 patients (65%) expressed their positivity and proactivity by practicing sports habitually (30%) or occasionally (35%).

The baseline characteristics of patients are extensively shown in Table 3.

## 5. Conclusions

Damoctocog alfa pegol appears to be a drug that can combine the needs of clinicians and patients, providing a high therapeutic efficacy profile, high adherence to therapy, and improved quality of life for treated subjects, all while reducing consumption. In the future, some new approaches could help us to better understand the topography [39], the nanomechanical properties [40], and the intermolecular interactions exerted by the protein receptors [41] of red blood cells to then compare the data obtained from healthy subjects and patients receiving prophylactic treatment with recombinant FVIII.

## Figures and Tables

**Figure 1 pharmaceuticals-16-01195-f001:**
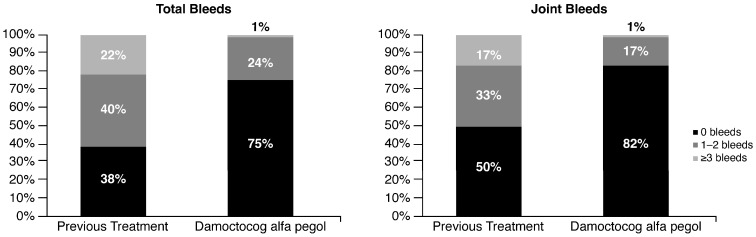
Comparison of the total bleeds and joint bleeds reported by 164 patients analyzed in the last 6 months of treatment with the previous FVIII concentrates before the switch to damoctocog alfa pegol (previous treatment) and in the last 6 months of treatment with damoctocog alfa pegol before the survey, respectively (damoctocog alfa pegol).

**Figure 2 pharmaceuticals-16-01195-f002:**
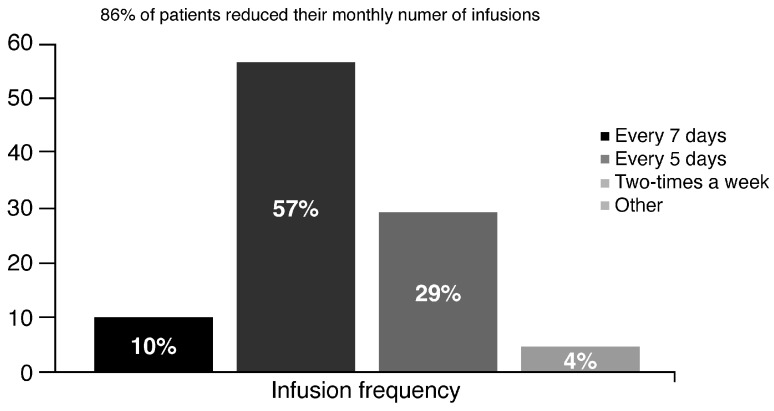
Frequency infusion with damoctocog alfa pegol.

**Table 1 pharmaceuticals-16-01195-t001:** Dosage and infusion time based on patient characteristics identified by the clinicians interviewed.

30–40 IU/kg Two-Times a Week	45–60 IU/kg Every 5 Days	60 IU/kg Every 7 Days
Patient characteristics: Active lifestyle (80%) 12–18 years (67%) Severe bleeding phenotype (67%) Playing sports (53%) Presence of synovitis (40%)	Patient characteristics: 19–65 years (60%) Active lifestyle (40%)	Patient characteristics: Over 65 (53%) Inactive or sedentary lifestyle (53%) Mild bleeding phenotype (53%) Poor autonomy (40%) Difficult venous access (33%) Comorbidities (27%)

In brackets, the percentage of clinicians indicating the specific characteristic.

**Table 2 pharmaceuticals-16-01195-t002:** Difference between the average monthly and annual dosage used in therapy with previous FVIII concentrate and damoctocog alfa pegol, expressed in IU/kg, and for a hemophilic patient weighing 70 kg (in grey).

	Previous FVIII Concentrate	Damoctocog Alfa Pegol Treatment	Difference (Δ)
IU/kg/month	389	321	−68
IU/kg/year	4673	3848	−825
IU/month (e.g., patient weight 70 kg)	27,230	22,470	−4760
IU/year (e.g., patient weight 70 kg)	326,760	269,640	−57,120

**Table 3 pharmaceuticals-16-01195-t003:** Baseline characteristics of patients in treatment with damoctocog alfa pegol.

Baseline Characteristics	n (%)
Patients	164 (100)
Age 12–18 years 19–40 years 41–65 years >65 years	25 (15) 63 (38) 60 (37) 16 (10)
Hemophilia A degree: Mild Moderate Severe	3 (2) 24 (15) 137 (83)
School or work activities: Full-time work Part-time work High school/university Retired No work/no school	71 (43) 24 (15) 40 (24) 16 (10) 13 (8)
Autonomy: Low Medium High	12 (7) 49 (30) 103 (63)
Comorbidities: Cardiovascular diseases Metabolic diseases Liver diseases Others	87 (53) 33 (20) 66 (40) 66 (40)
Sports activity: None Occasionally Habitually	58 (35) 57 (35) 49 (30)
Attitude: Pessimistic/disheartened Unresponsive/indifferent Proactive/seeking continuous improvement	12 (7) 22 (13) 130 (79)

## Data Availability

Data is contained within the article.

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
