# Peer review of "Damoctocog Alfa Pegol for Hemophilia A Prophylaxis: An Italian Multicenter Survey"

_pharmaceuticals, 2023, doi:10.3390/ph16091195_

Round 1

Reviewer 1 Report

Congratulations for your paper!!!

Only two remarks

In section 2.2 you mention the switch to damoctocog, but you don't explain the reason and I think it would be really interesting

In section 2.3 clinicians' perception is explained. I think it should be explained the methods (Questions of the questionnaire) and results from the questions otherwise what this section explains seems more an opinion than a fact

Author Response

REVIEWER 1

Congratulations for your paper!!!

Authors’ response:

Many thanks for your interest in the study and for your kind words.

Only two remarks

In section 2.2 you mention the switch to damoctocog, but you don't explain the reason and I think it would be really interesting

Authors’ response:

Thank you for this suggestion. The reason for the switch to damoctocog has been added in the revised manuscript (see lines 120-128)

In section 2.3 clinicians' perception is explained. I think it should be explained the methods (Questions of the questionnaire) and results from the questions otherwise what this section explains seems more an opinion than a fact

Authors’ response:

Thank you for this suggestion. Methods and results from the questions have been added in the revised manuscript (see lines 193-198).

Reviewer 2 Report

The manuscript titled “Damoctocog alfa pegol for hemophilia A prophylaxis: an Italian multicenter survey” is an original work where the author established a comparative study of hemopholic A patients treated with damoctocog alfa pegol and other previous treatments. After 6 months of exposure, it was observed a greatly amelioration of these patients in terms of bleeds, hemarthroses and the amount of monthly infusions. The scientific content is interesting and the sections are well-designed.

However, it exists some points that need to be addressed (please, see them below detailed point-by-point). The most relevant outcomes found by the authors can contribute to in the growth of many fields like the clinical&healthcare, specially to design the next-generation of therapies against hemophilia A. For this reason, I will recommend the present scientific manuscript for further publication in Pharmaceuticals once all the below described suggestions will be properly fixed.

Here, there exists some points that must be covered in order to improve the scientific quality of the manuscript paper:

1) INTRODUCTION. “In recent years, alongside (…)” (lines 64-65). What is the worldwide prevalance of hemophilia A? The author should state that 400,000 people are estimated to suffer hemophilia. Meta-analysis studies unravel that nearby 1,125,000 men have the inherited bleeding disorder which 418,000 have a sever version of this disease [1].

[1] Iorio, A.; et al. Establishing the Prevalence and Prevalence at Birth of Hemophilia in Males: A Meta-analytic Approach Using National Registries. Ann. Intern. Med. 2019, 171, 540-546. https://doi.org/10.7326/M19-1208.

2) “The efficacy and safety of this drug (…) PROTECT VIII study” (lines 72-74). Even if I completely agree with this statement provided by the author, it would be also necceseary to point out the adverse reactions and contraindications observed when the damoctocog alfa pegol is prescribed to hemophilic A patients. There exists a broad range of side effects like headache (13%), cough (8%), nausea (4%) or pyrexia (9%) [2].

[2] Jivi (antihemophilic factor-recombinant pegylated-aucl kit). DailyMed. 2018. Retrieved 1 October 2020.

3) RESULTS. “2.1. Patients treated with damoctocog alfa pegol: characteristics” (lines 107-121). The author should consider to move all this content to the “Methods” section (Please, see the point indicated below in this regard).

4) “2.2. Switch to damoctocog alfa pegol. A total of 86% of patients currently receiving (…) treated in prophylaxis with other FVIII concentrates” (lines 122-124). A schematic representation of how the drug interacts with the Factor VIII and induces its activation and subsequent coagulation cascade effect for the proteolytic activation of factor X by factor IXa could significantly benefit the comprehension of the readers to better understand the key role of damoctocog alfa pegol in patients with hemophilia A.

5) “Almost all patients (83%) in the previous prophylactic treatment used (…) (another octocog alfa, turoctocog alfa, moroctocog alfa, rFVIII-single chain, rhFVIII)” (lines 126-129). Please, the author should provide more details about the used previous treatments by the hemophilic patients (e.g. dosage, time exposure, …).

6) DISCUSSION. The most relevant outcomes found by the author are clearly stated in this section. No further actions are requested.

7) METHODS. (OPTIONAL) The author should consider to change the current name of this section by “Materials and methods”. Then, the author should specify here not only the number of patients, but also their race, sex and age. Moreover, the manufacturer/supplier details should be furnished for all the techniques, chemicals and consumables used in this study. Finally, the therapy dose (IU/kg) of damoctocog alfa pegol must be also indicated in this section (data coming from Table 2, line 162).

8) CONCLUSION. This section is clear and concise. Nevertheless, it may be appropiate if the author could highlight some future avenues to pursue this research and the potential benefits this study could have for the society. In this context, the author need to mention the use of single molecule techniques to investigate the topography [3], the nanomechanical properties [4] and the intermolecular interactions exerted by the protein receptors [5] of red blood cells and compare these data in healthy patients and patients with hemophilia A following the treatment with damoctocog alfa pegol.

[3] Sergunova, V.; et al. Investigation of Red Blood Cells by Atomic Force Microscopy. Sensors 2022, 22, 2055. https://doi.org/10.3390/s22052055

[4] Magazzù, A.; et al. Investigation of Soft Matter Nanomechanics by Atomic Force Microscopy and Optical Tweezers: A comprehensive Review. Nanomaterials 2023, 13, 963. https://doi.org/10.3390/nano13060963.

[5] Lostao, A.; et al. Recent advances in sensing the inter-biomolecular interactions at the nanoscale – A comprehensive review of AFM-based force spectroscopy. Int. J. Biol. Macromol. 2023, 238, 124089. https://doi.org/10.1016/j.ijbiomac.2023.124089.

9) REFERENCES. The references are in the proper format style of Pharmaceuticals. No actions are requested from the author.

Minor editing of English language required.

Author Response

I will recommend the present scientific manuscript for further publication in Pharmaceuticals once all the below described suggestions will be properly fixed.

Authors’ response:

Thank you for your interest in the manuscript, I greatly appreciated your careful revision.

Here, there exists some points that must be covered in order to improve the scientific quality of the manuscript paper:

1) INTRODUCTION. “In recent years, alongside (…)” (lines 64-65). What is the worldwide prevalance of hemophilia A? The author should state that 400,000 people are estimated to suffer hemophilia. Meta-analysis studies unravel that nearby 1,125,000 men have the inherited bleeding disorder which 418,000 have a sever version of this disease [1].

[1] Iorio, A.; et al. Establishing the Prevalence and Prevalence at Birth of Hemophilia in Males: A Meta-analytic Approach Using National Registries. Ann. Intern. Med2019171, 540-546. https://doi.org/10.7326/M19-1208.

Authors’ response:

Thank you for this suggestion. The introduction has been revised accordingly (see lines 70-75).

2) “The efficacy and safety of this drug (…) PROTECT VIII study” (lines 72-74). Even if I completely agree with this statement provided by the author, it would be also necceseary to point out the adverse reactions and contraindications observed when the damoctocog alfa pegol is prescribed to hemophilic A patients. There exists a broad range of side effects like headache (13%), cough (8%), nausea (4%) or pyrexia (9%) [2].

[2] Jivi (antihemophilic factor-recombinant pegylated-aucl kit). DailyMed. 2018. Retrieved 1 October 2020.

Authors’ response:

Thank you for this observation. The manuscript has been revised accordingly (see lines 87-90).

3) RESULTS. “2.1. Patients treated with damoctocog alfa pegol: characteristics” (lines 107-121). The author should consider to move all this content to the “Methods” section (Please, see the point indicated below in this regard).

Authors’ response:

Thank you for this suggestion. Accordingly, this section has been moved to Methods.

4) “2.2. Switch to damoctocog alfa pegol. A total of 86% of patients currently receiving (…) treated in prophylaxis with other FVIII concentrates” (lines 122-124). A schematic representation of how the drug interacts with the Factor VIII and induces its activation and subsequent coagulation cascade effect for the proteolytic activation of factor X by factor IXa could significantly benefit the comprehension of the readers to better understand the key role of damoctocog alfa pegol in patients with hemophilia A. 

Authors’ response:

Thank you for this suggestion. With respect, I believe that the addition of the proposed figure will not add value to the reader with specific regard to the drug interaction with FVIII. Indeed, damoctocog alfa pegol carries out its action like the endogenous FVIII, therefore the figure should represent the classical coagulation cascade. Pegylation is only a method to prolong the permanence of exogenous FVIII in the circulation, but does not intervene in the coagulation cascade. I hope that you agree with my thinking.

5) “Almost all patients (83%) in the previous prophylactic treatment used (…) (another octocog alfa, turoctocog alfa, moroctocog alfa, rFVIII-single chain, rhFVIII)” (lines 126-129). Please, the author should provide more details about the used previous treatments by the hemophilic patients (e.g. dosage, time exposure, …).

Authors’ response:

Thank you for this remark. Accordingly, dosage and time exposure of previous treatments have been added in the revised manuscript (see lines 138-140).

6) DISCUSSION. The most relevant outcomes found by the author are clearly stated in this section. No further actions are requested.

Authors’ response:

Thank you.

7) METHODS. (OPTIONAL) The author should consider to change the current name of this section by “Materials and methods”. Then, the author should specify here not only the number of patients, but also their race, sex and age. Moreover, the manufacturer/supplier details should be furnished for all the techniques, chemicals and consumables used in this study. Finally, the therapy dose (IU/kg) of damoctocog alfa pegol must be also indicated in this section (data coming from Table 2, line 162).

Authors’ response:

Thank you for this suggestion. The title of the section and text have been revised accordingly.

8) CONCLUSION. This section is clear and concise. Nevertheless, it may be appropiate if the author could highlight some future avenues to pursue this research and the potential benefits this study could have for the society. In this context, the author need to mention the use of single molecule techniques to investigate the topography [3], the nanomechanical properties [4] and the intermolecular interactions exerted by the protein receptors [5] of red blood cells and compare these data in healthy patients and patients with hemophilia A following the treatment with damoctocog alfa pegol.

[3] Sergunova, V.; et al. Investigation of Red Blood Cells by Atomic Force Microscopy. Sensors 202222, 2055. https://doi.org/10.3390/s22052055

[4] Magazzù, A.; et al. Investigation of Soft Matter Nanomechanics by Atomic Force Microscopy and Optical Tweezers: A comprehensive Review. Nanomaterials 202313, 963. https://doi.org/10.3390/nano13060963.

[5] Lostao, A.; et al. Recent advances in sensing the inter-biomolecular interactions at the nanoscale – A comprehensive review of AFM-based force spectroscopy. Int. J. Biol. Macromol2023238, 124089. https://doi.org/10.1016/j.ijbiomac.2023.124089.

Authors’ response:

Thank you for this suggestion. The conclusion has been revised accordingly (see lines 336-340).

9) REFERENCES. The references are in the proper format style of Pharmaceuticals. No actions are requested from the author.

Authors’ response:

Thank you.

Round 2

Reviewer 2 Report

The author has done a great deal of effort and the scientific manuscript content has subsequently been improved.

For this reason, I warmly recommend this work for further publication in Pharmaceuticals.

The manuscript is well-written.